# Impact of COVID-19 vaccination on mortality after acute myocardial infarction

**Mohit D. Gupta** [1]ᵒ*, **Shekhar Kunal**[2]ᵒ, **Girish M. P.**[1], **Dixit Goyal**[1], **Rajeev Kumar Malhotra**[3], **Prashant Mishra**[1], **Mansavi Shukla**[1], **Aarti Gupta**[1], **Vanshika Kohli**[1], **Nitya Bundela**[1], **Vishal Batra**[1], **Ankit Bansal**[1], **Rakesh Yadav**[4], **Jamal Yusuf**[1], **Saibal Mukhopadhyay**[1]

1 Department of Cardiology, Govind Ballabh Pant Institute of Post Graduate Medical Education and Research, Delhi, India, 2 Department of Cardiology, ESIC Medical College and Hospital, Faridabad, Haryana, 3 Delhi Cancer Registry, Institute Rotary Cancer Hospital, All India Institute of Medical Sciences, New Delhi, India, 4 Department of Cardiology, All India Institute of Medical Sciences, New Delhi, India

ᵒ These authors contributed equally to this work.
* drmohitgupta@yahoo.com

## Abstract

### Background

COVID-19 vaccines are highly immunogenic but cardiovascular effects of these vaccines have not been properly elucidated.

### Objectives

To determine impact of COVID-19 vaccination on mortality following acute myocardial infarction (AMI).

### Methods

This was a single center retrospective observation study among patients with AMI enrolled in the the North India ST-Elevation Myocardial Infarction (NORIN-STEMI) registry. In all the enrolled patients, data regarding patient's vaccination status including details on type of vaccine, date of vaccination and adverse effects were obtained. All enrolled subjects were followed up for a period of six months. The primary outcome of the study was all-cause mortality both at one month and at six months of follow-up. Propensity-weighted score logistic regression model using inverse probability of treatment weighting was used to determine the impact of vaccination status on all-cause mortality.

### Results

A total of 1578 subjects were enrolled in the study of whom 1086(68.8%) were vaccinated against COVID-19 while 492(31.2%) were unvaccinated. Analysis of the temporal trends of occurrence of AMI post vaccination did not show a specific clustering of AMI at any particular time. On 30-day follow-up, all-cause mortality occurred in 201(12.7%) patients with adjusted odds of mortality being significantly lower in vaccinated group (adjusted odds ratio [aOR]: 0.58, 95% CI: 0.47–0.71). Similarly, at six months of follow-up, vaccinated AMI group

**Data Availability Statement:** Data cannot be shared publicly because of ethical reasons. The Data includes potentially identifiable /sensitive information on covid vaccination details of patients that cannot be made available on public domain as

per government rules and regulations of the country. However, data are available on request on drmohitgupta@Yahoo.com or iecmamc@gmail.com (Ethics Committee) for researchers who meet the criteria for access to confidential data.

**Funding:** The author(s) received no specific funding for this work.

**Competing interests:** The authors have declared that no competing interests exist.

had lower odds of mortality(aOR: 0.54, 95% CI: 0.44 to 0.65) as compared to non-vaccinated group.

## Conclusions

COVID-19 vaccines have shown to decrease all-cause mortality at 30 days and six months following AMI.

## Introduction

The unprecedented emergence of COVID-19 pandemic saw accelerated development of treatment strategies including drugs and vaccines [1]. Most of the vaccines were developed in a shorter time frame however, still had high efficacy and safety. In India, "Emergency use authorization" was granted to two vaccines viz. COVISHIELD (Serum Institute of India Limited, India) and COVAXIN (Bharat Biotech Limited, India) [2]. The universal vaccination against COVID-19 in India began in January 2021 and within a span of 2 years, 2.2 billion doses of COVID-19 vaccine (COVISHIELD: 79.3%; COVAXIN: 16.5%) has been administered [3]. Several large-scale studies and clinical trials have established the efficacy and safety of COVID-19 vaccines. However, most of these studies and trials were conducted in ideal settings among homogenous and limited population groups far away from the real-world scenario [4]. The adverse effects (AEs) of COVID-19 vaccine have mostly been mild, transient and self-limiting. However, concerns have been raised regarding the cardiovascular adverse effects of these vaccines. Any side effect can have catastrophic effect especially in large densely populated countries such as India. Data from the COVID-19 vaccination trials reported that adverse cardiovascular effects were largely isolated with an incidence of <0.05% [5, 6]. However, recent real-world data have highlighted increasing frequency of cardiovascular AEs with COVID-19 vaccines. A majority of these AEs were cases of myocarditis caused by mRNA-based vaccines [7, 8]. With increasing concerns regarding cardiovascular AEs especially acute coronary syndrome (ACS), there is a greater hesitancy to get vaccinated thereby affecting vaccination rates and prolonging the COVID-19 pandemic. This is of greater concern especially with emergence of newer potent variants of SARS-CoV-2 which have been associated with greater reinfection rates and worse outcomes. Previous studies among patients with heart failure have shown that COVID-19 vaccination was associated with significant reduction in all-cause hospitalization rates and mortality [9]. However, data regarding occurrence and impact of acute myocardial infarction (AMI) following COVID-19 vaccination is sparse. We studied timeline of occurrence of AMI after COVID-19 vaccination and its impact on all-cause mortality in these patients.

## Methods

This was a single center retrospective observational study conducted in the Department of Cardiology at a tertiary care medical center. All patients ≥18 years of age with ST-Elevation Myocardial Infarction (STEMI) who were registered in the North India ST-Elevation Myocardial Infarction (NORIN-STEMI) registry [10] and had consented to be a part of the study were enrolled from August 2021 up till August 2022. In all these patients, STEMI was diagnosed based on the European Society of Cardiology/American College of Cardiology Foundation/American Heart Association/World Heart Federation Task Force for the Fourth Universal Definition of Myocardial Infarction [11]. In all the enrolled patients following a written

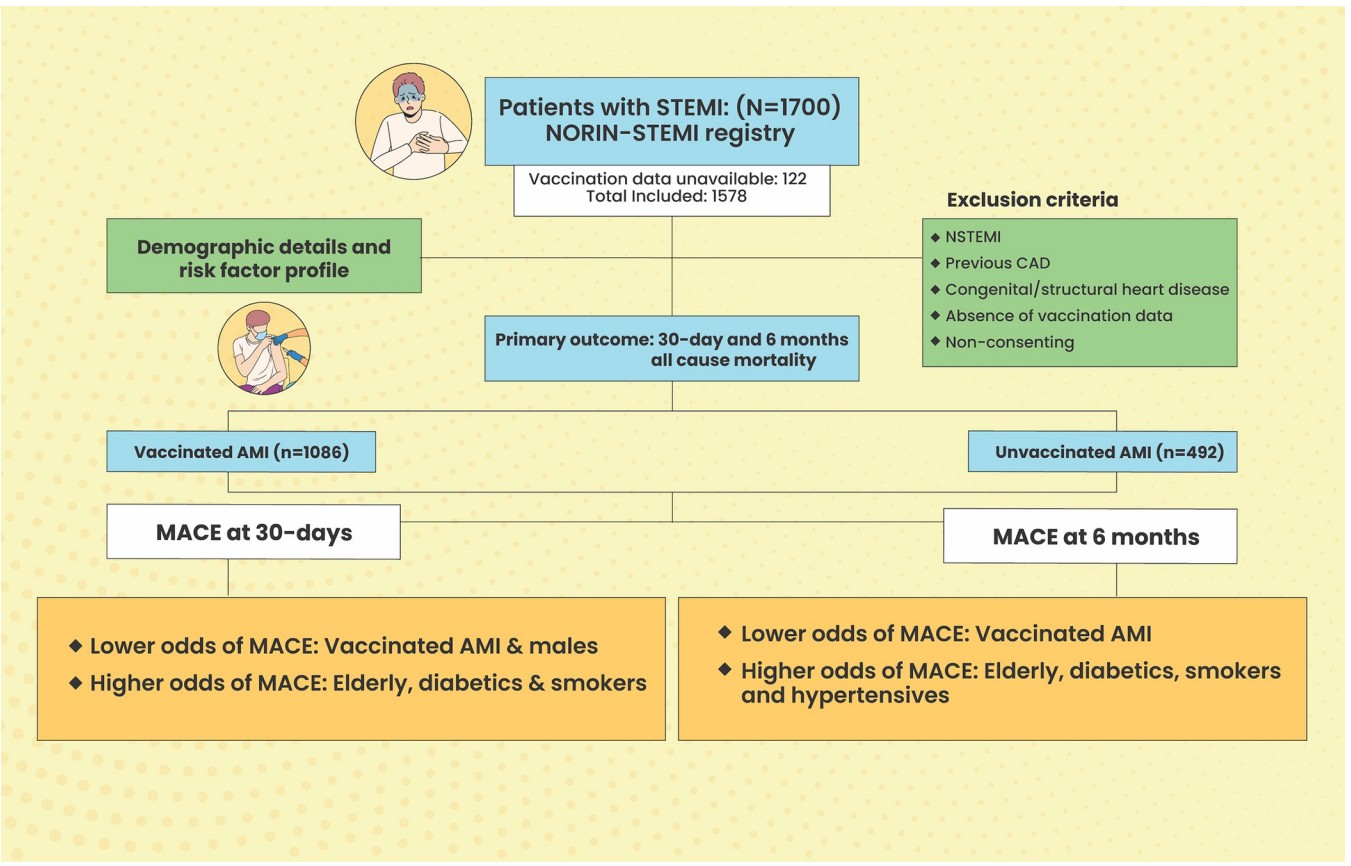

**Fig 1. Central figure highlighting the impact of vaccination on all-cause mortality in acute myocardial infarction (AMI) patients.** Abbreviations: NORIN-STEMI: North Indian ST-Segment Elevation Myocardial Infarction; STEMI: ST-Segment Elevation Myocardial Infarction; MACE: major adverse cardiac events; CAD: coronary artery disease; NSTEMI: non ST-Segment Elevation Myocardial Infarction.

informed consent, baseline demographic data, risk factors and presenting symptoms were recorded in an electronic questionnaire. The details of the treatment given were also recorded. Data regarding the patient's vaccination status against COVID-19 which included details on the type of vaccine, date of vaccination (first dose and second dose) and adverse effect following immunization (AEFI). Patients with non-ST-elevation myocardial infarction (NSTEMI), history of previous MI or coronary revascularization, congenital heart disease, previous structural heart disease, recent surgery, absence of vaccination details, unwilling for follow-up or non-consenting to be a part of the study were excluded. Subjects were labelled as vaccinated if they had received either one or both the doses of COVID-19 vaccine while unvaccinated population included those who had not received a single dose of the vaccine (Fig 1). In order to determine the temporal association between vaccination and occurrence of AMI, time-lines were drawn which included 0–30 days, 30–90 days, 90–150 days, 150–210 days, 210–270 days, 270–330 days and beyond.

## Outcomes

The primary outcome of the study was major adverse cardiac events (MACE) in terms of all-cause mortality both at one month and at six months of follow-up. Outcome data were collected from the medical records and via telephonic interview.

### Consent and ethical issues

A written informed consent was obtained from all the enrolled patients. The present study was approved by the institutional ethics committee [EC number: F.1/IEC/MAMC/ (85/04/2021/ No 486)] and was conducted in accordance with the ethical principles as per the Declaration of Helsinki and are consistent with Good Clinical Practice and all local regulations.

### Statistics

Continuous data was expressed as median and categorical data as proportions. Normality of distribution of continuous variables were assessed using the Kolmogorov-Smirnov test. Chi-square test was used for unweighted comparison. In order to account for differences in the baseline patients' characteristics, a propensity-weighted score logistic model using the inverse probability of treatment weighting (IPTW) was used. The propensity score (PS) was determined using logistic regression with vaccination as the dependent variable and age, sex, diabetes, hypertension, and smoking as covariates. The vaccinated individuals were weighted with 1/PS and nonvaccinated patients with 1/(1-PS). The imbalance of covariates before and after IPTW was assessed with P-value and considered balanced if P<0.05. A two-sided P value of < 0.05 was considered to be statistically significant. SPSS version 24.0 (IBM Corp, Armonk, NY) and R-software was used for statistical analysis.

## Results

A total of 1700 patients with AMI were initially screened of whom vaccination data was not available for 122 subjects and they were excluded from the study (Fig 1). Of the 1578 subjects enrolled in the study, a majority of them were males (1273;80.7%) with a median age of 55 years (IQR: 46–62 years). A total of 1086 patients (68.8%) were vaccinated against COVID-19 while 492 subjects (31.2%) had not received a single dose of COVID-19 vaccine and were labelled as unvaccinated. Among the vaccinated group, 1047 (96.4%) had received two doses of the vaccine while 39 (3.6%) had received only a single dose. Majority of them (1002 [92.3%]) had been vaccinated with COVISHIELD while 84 (7.7%) had received COVAXIN. The demographics and baseline characteristics in the vaccinated and unvaccinated group were comparable (Table 1). Analysis of the temporal trends of occurrence of AMI following vaccination did not show a specific clustering of AMI after the vaccination at any particular time. The numbers of events according to the timeline are shown in Fig 2. A total of 185 (11.7%) of STEMIs occurred within 90–150 days of vaccination while 175 (11.1%) occurred between 150–270 days. Only 28 (1.8%) of AMI cases occurred within first 30 days of COVID-19 vaccination.

### Outcomes

**(i) 30-day MACE.** Of the 1578 patients with AMI, the 30-day all-cause mortality occurred in 201(12.7%) patients (S1 Table). Out of these, 116(57.7%) belonged to the vaccinated group while 85 (42.2%) were non-vaccinated. The adjusted odds of 30-day mortality was significantly lower in the vaccinated AMI population (adjusted odds ratio[aOR]: 0.58, 95% CI: 0.47–0.71) as compared to the non-vaccinated group. Male patients with AMI had a lower likelihood of 30-day mortality as compared to women (OR: 0.41; 95% CI:0.31–0.53). Contrarily, increasing age, diabetics and smokers had higher odds of 30-day mortality (Fig 3A). A lower risk of 30-day mortality in the vaccinated AMI patients was observed in all the sub-groups, albeit some of the factors did not reach statistical significance [Table 2].

**(ii) MACE between 30-days and six months.** During the period of 30 days to six months, 75 patients had all-cause mortality of whom 43.7% were vaccinated. The adjusted odds of

**Table 1. Baseline patients characteristics of vaccinated and unvaccinated AMI patients.**

| Variable | Vaccinated (n = 1086) | Unvaccinated (n = 492) | P-value (Unweighted) | P-value (Weighted) |
|---|---|---|---|---|
| **Sex** | | | | |
| Male | 895(82.4%) | 378(76.8%) | 0.009 | 0.944 |
| Female | 191(17.6%) | 114(23.2%) | | |
| Age, Median(IQR), yrs | 55[46–62] | 56[48–65] | <0.001 | 0.939 |
| 18–39 yrs | 112(40.3%) | 33(6.7%) | | |
| 40–64 yrs | 760(70.0%) | 325(66.1%) | | |
| ≥65 yrs | 214(19.7%) | 134(27.2%) | | |
| **Co-morbidities** | | | | |
| Diabetes | 233(21.5%) | 114(23.2%) | 0.446 | 0.924 |
| Hypertension | 347(32.0%) | 156(31.7%) | 0.923 | 0.992 |
| COPD | 35 (3.2%) | 15 (3.0%) | 0.854 | |
| CKD | 03 (0.27%) | 01 (0.20%) | 0.789 | |
| Heart failure | 30 (2.7%) | 18 (3.6%) | 0.336 | |
| Dyslipidaemia | 42(3.9%) | 24(4.9%) | 0.353 | 0.961 |
| Smoking | 474(43.6%) | 184(37.4%) | 0.020 | 0.958 |
| Physical Activity | 484(44.6%) | 217(44.1%) | 0.864 | |
| Family History | 261(24.0%) | 120(21.4%) | 0.878 | |
| Revascularization | 52 (10.6%) | 139 (12.1%) | 0.208 | 0.894 |

**Abbreviations:** AMI: acute myocardial infarction; COPD: chronic obstructive pulmonary disease; CKD: chronic kidney disease; IQR: inter-quartile range; yrs: years

mortality in vaccinated subjects were lower than in the non-vaccinated (aOR: 0.34[0.24–0.48]) [Fig 3B]. Apart from vaccination, increasing age was also significantly associated with higher mortality [S2 Table].

**(iii) MACE within six months.** Over a period of six months of follow-up, 276 (12.7%) died within six months of the admission to hospital, of whom 148(53.6%) were vaccinated with either dose while 128(46.4%) were non-vaccinated. The adjusted odds of mortality in the first six months were significantly lower in the vaccinated AMI group (aOR: 0.54, 95% CI: 0.44 to 0.65) as compared to the non-vaccinated group. Additionally, males patients with AMI had a lower likelihood of mortality (aOR: 0.45; 95% CI: 0.35–0.58) within six months as compared to females. On the other hand, increasing age, diabetes, hypertension, and smoking increased the odds of six months mortality (Fig 3C). A lower risk of six months mortality in vaccinated AMI patients was observed in all the sub-groups, albeit some of the factors did not reach statistical significance [Table 2].

## Discussion

The present study retrospectively investigated the impact of vaccination on occurrence of AMI and all-cause mortality at 30 days and six months. Findings of our study showed that the 30-day and six months all-cause mortality risk was significantly lower in the vaccinated subjects as compared to the unvaccinated population. Additionally, a lack of temporal clustering of AMI cases following COVID-19 vaccination did not suggest a significant association between COVID-19 vaccination and occurrence of AMI. This study is the first to be conducted among a larger population of AMI patients which has shown COVID-19 vaccine to be not only safe but also have a protective effect in terms of reduction of all-cause mortality both on short term as well as at six months of follow-up.

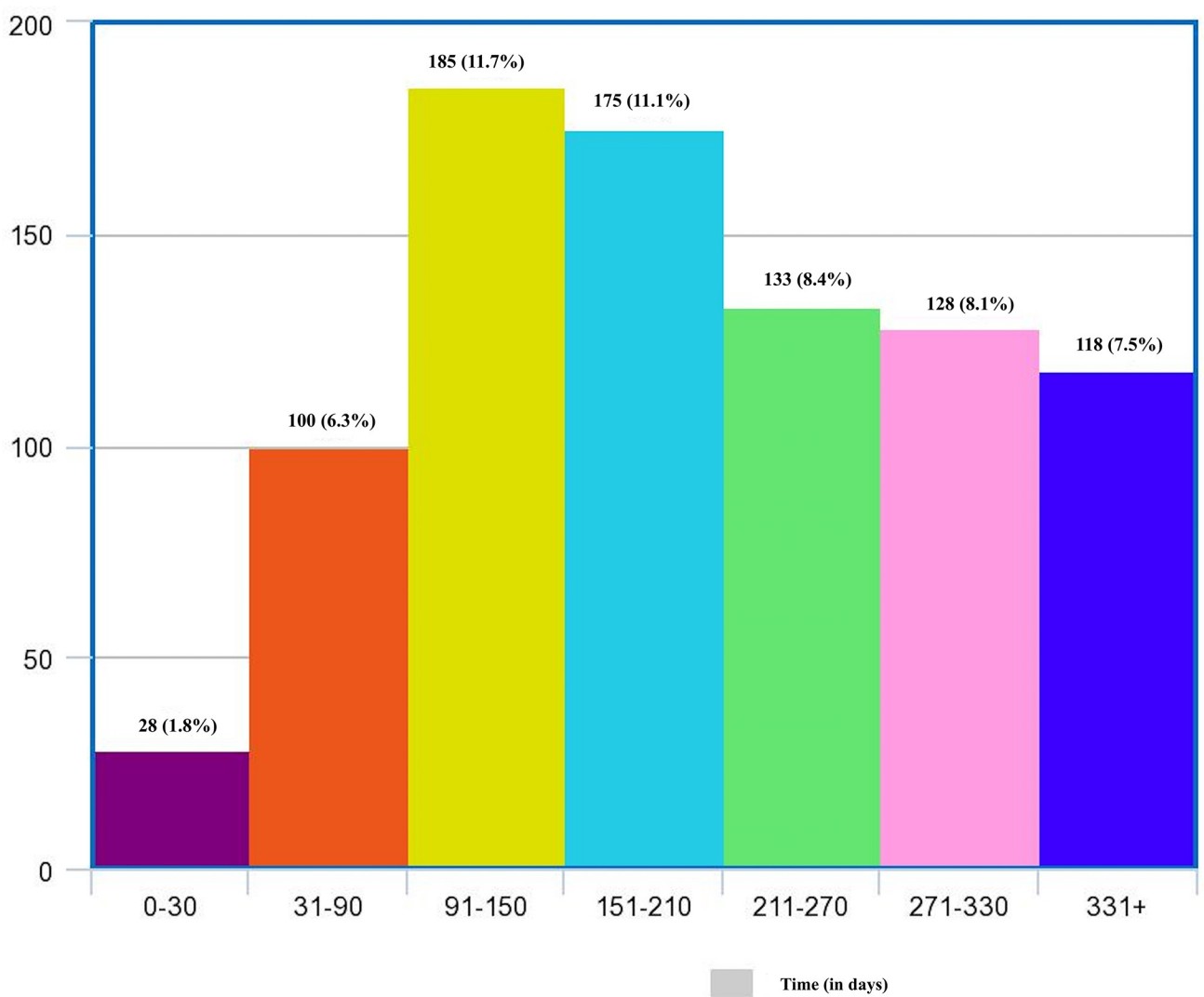

**Fig 2. Histogram showing the timeline for distribution of AMI patients following COVID-19 vaccination.**

COVID-19 infection has been associated with multiple CV events including acute coronary syndrome (ACS), myocarditis and heart failure [12]. CV effects of COVID-19 infection depends both on the disease severity as well as the presence of co-morbidities. The two COVID-19 vaccines used in India have been shown to generate high neutralizing antibody titres and stronger T-cell response [13]. Data regarding the protective effects of COVID-19 vaccine on CV system is largely limited to a small unpublished study [14]. It reported that COVID-19 vaccination was associated with decreased rates of hospital admission ACS patients undergoing percutaneous coronary intervention. The present study also showed significant reduction over and above the other risk factors in all-cause mortality in vaccinated AMI patients at 30 days and at six months of follow-up.

The exact mechanism regarding the beneficial effects of COVID-19 vaccination in reduction of MACE is unclear. A recent study among patients with heart failure demonstrated protective effects of COVID-19 vaccination in terms of significant reduction in all-cause mortality (HR 0.33; CI 0.23–0.48) on follow-up [9]. Viral infections such as influenza has been shown to

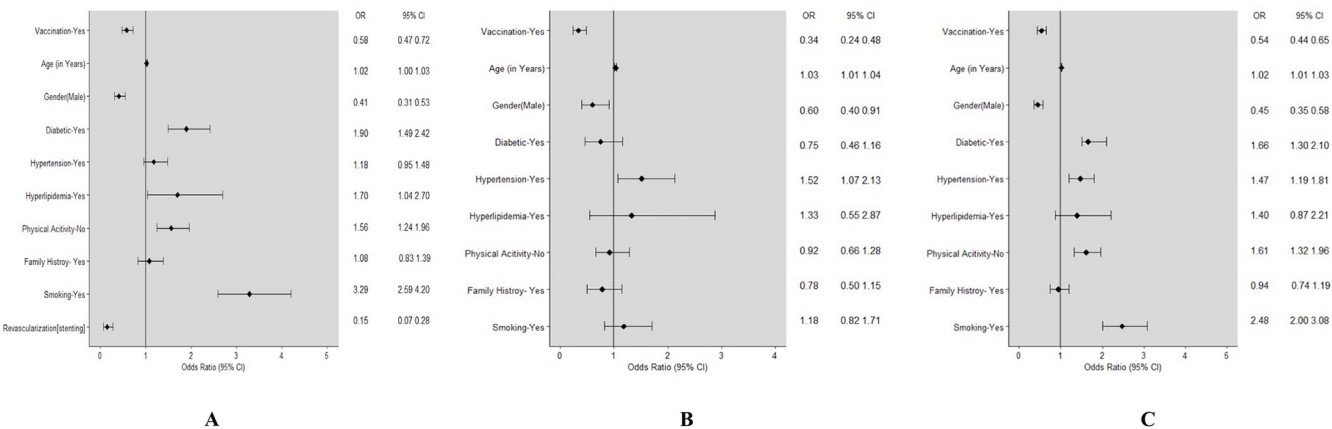

**Fig 3. A**: Forest plot showing the adjusted odds ratio for the factors for 30-days all-cause mortality using the propensity score weighting. **B**: Forest plot showing the adjusted odds ratio for all-cause mortality between 30-days and six months using the propensity score weighting. **C**: Forest plot showing the adjusted odds ratio for the factors all-cause mortality using the propensity score weighting at six months.

increase the predisposition for CV events [15]. This has been attributed to a heightened systemic inflammatory response following infection leading to plaque rupture [15]. Previous studies based on the influenza vaccines have shown to reduce MACE in patients with ACS [16–18]. The proposed hypothesis regarding protective effect of influenza vaccine is the cross-reaction of the vaccine-induced antibodies with the bradykinin receptor thereby leading to increased nitric oxide production and vasodilation [19]. A similar analogy can be thought in terms of the COVID-19 vaccine however, greater research is needed to explore this aspect. Another plausible reason regarding the beneficial effects of COVID-19 vaccination in terms of reduction in all-cause mortality following AMI could be the "healthy user effect" [20]. Individuals who undergo COVID-19 vaccination can be thought of as "healthy users" who are inclined to adopt behaviors that promote their well-being, such as making dietary and lifestyle modifications, and adhering to medication regimens following AMI thereby leading to lower mortality as compared to unvaccinated individuals.

Additionally, vaccination against COVID-19 has been shown to reduce the risk of occurrence of AMI and stroke following COVID-19. Data from the Korean nationwide COVID-19 registry and the Korean National Health Insurance Service involving 231037 patients reported that the adjusted risk for both AMI (aHR: 0.48; 95% CI, 0.25–0.94) and ischemic stroke (aHR: 0.40; 95% CI, 0.26–0.63) following COVID-19 infection was significantly lower in fully vaccinated patients [21]. Similarly, data from the United States National COVID Cohort Collaborative reported that both full (aHR: 0.59; 95% CI: 0.55–0.63) and partial (aHR: 0.76; 95% CI: 0.65–0.89) vaccination against COVID-19 were associated with reduced risk of MACE [22].

COVID-19 vaccination was initially implicated in causing AMI without any definitive evidence. This was based on a few case reports [23–26] rather than any systematically conducted study. Few case reports and series did initially ascribe the role of COVID-19 vaccination to the occurrence of AMI [23–26]. Multiple mechanisms that were postulated included (a) vaccine induced thrombocytopenia [27], (b) Kounis syndrome as a part of allergic vasospasm to the components of COVID-19 vaccine [28], (c) myocarditis, (d) stress of obtaining a vaccine among elderly individuals with pre-existing health conditions resulting in demand-supply mismatch ischemia [29]. However, recent studies [30, 31] have failed to establish a definite association of AMI with COVID-19 vaccine. Post-mortem examination and histopathological evaluation of five sudden cardiac deaths following COVID-19 vaccination reported no definitive causal relationship with vaccine administration [30]. Similarly, in a nationwide study from

**Table 2. Propensity score weighted logistic regression models for assessing odds ratio of vaccination on 30 days and 6 months mortality.**

| Variable | 30-day mortality (n = 201) | | | | 6-Month Mortality(n = 276) | | | |
|---|---|---|---|---|---|---|---|---|
| | Vaccinated (n = 116) | Unvaccinated (n = 85) | Adjusted Odds Ratio (95% CI) | P-value | Vaccinated (n = 148) | Unvaccinated (n = 128) | Adjusted Odds Ratio (95% CI) | P-value |
| Overall* | 116 | 85 | 0.58 [0.46–0.71] | <**0.001** | 148 | 128 | 0.54 [0.44–0.66] | <**0.001** |
| Sub-group Analysis* | | | | | | | | |
| Male | 88 | 57 | 0.60 [0.49–0.79] | <**0.001** | 116 | 84 | 0.62 [0.50–0.78] | <**0.001** |
| Female | 28 | 28 | 0.41 [0.25–0.66] | **0.0003** | 32 | 44 | 0.30 [0.19–0.47] | <**0.001** |
| Age* | | | | | | | | |
| 18–39 years | 8 | 5 | 0.36 [0.14–0.82] | **0.0198** | 10 | 6 | 0.49 [0.22–1.08] | 0.080 |
| 40–64 years | 80 | 48 | 0.68 [0.52–0.88] | **0.0039** | 101 | 73 | 0.64 [0.50–0.82] | **0.0003** |
| ≥65 years | 28 | 32 | 0.39 [0.25–0.61] | <**0.001** | 37 | 49 | 0.38 [0.26–0.56] | <**0.001** |
| Diabetes | | | | | | | | |
| Yes | 38 | 31 | 0.48 [0.32–0.71] | **0.0003** | 42 | 39 | 0.47 [0.31–0.71] | **0.0004** |
| No | 78 | 54 | 0.60 [0.47–0.78] | **0.0001** | 106 | 89 | 0.54 [0.43–0.68] | <**0.001** |
| Hypertension | | | | | | | | |
| Yes | 50 | 34 | 0.62 [0.44–0.87] | **0.0007** | 60 | 51 | 0.62 [0.44–0.86] | <**0.001** |
| No | 66 | 51 | 0.56 [0.42–0.74] | <**0.001** | 88 | 77 | 0.51 [0.40–0.65] | <**0.001** |
| Hyperlipidaemia | | | | | | | | |
| Yes | 8 | 6 | 1.58 [0.58–4.53] | 0.380 | 8 | 9 | 0.61 [0.22–1.69] | 0.339 |
| No | 108 | 79 | 0.56 [0.45–0.70] | <**0.001** | 140 | 119 | 0.54 [0.44–0.66] | <**0.001** |
| Smoking | | | | | | | | |
| Smoker | 72 | 47 | 0.55 [0.41–0.75] | **0.0002** | 83 | 61 | 0.54 [0.40–0.73] | <**0.001** |
| Non-smoker | 44 | 38 | 0.57 [0.42–0.79] | **0.0008** | 65 | 67 | 0.52 [0.40–0.69] | <**0.001** |
| Physical Activity | | | | | | | | |
| Yes | 43 | 23 | 0.85 [0.59–1.22] | 0.3789 | 65 | 41 | 0.64 [0.47–0.86] | **0.004** |
| No | 73 | 62 | 0.49 [0.37–0.64] | <**0.001** | 83 | 87 | 0.48 [0.37–0.63] | <**0.001** |
| Family History | | | | | | | | |
| Yes | 27 | 20 | 0.54 [0.3–0.84] | **0.007** | 37 | 26 | 0.82 [0.53–1.24] | 0.345 |
| No | 89 | 65 | 0.58 [0.45–0.74] | <**0.001** | 111 | 102 | 0.48 [0.38–0.60] | <**0.001** |

*Adjusted for age, sex, comorbidities, smoking, physical activity, and family history (in subgroup analysis respective variable was removed from the model)

France in individuals ≥75 years, there was no increase in the incidence of AMI 14 days following administration of each dose of the BNT162b2 (mRNA) vaccine [31]. The present study failed to show any temporal clustering of cases of AMI following COVID-19 vaccination.

## Limitations

This was a single center retrospective study. This is a main limitation. The findings need to be validated in further larger studies from different ethnic groups. However, as evident by the high immunogenic response generated by these two vaccines shown in different populations across the country, it is likely that the vaccine will show a protective effect. In our study, only all-cause mortality was evaluated on follow-up which is also one of the limitations. All-cause mortality is a hard end point and we have tried to adjust all the possible confounders in our analysis. In this study during covid times, it was extremely challenging to collect data of all possible outcomes in a resource limited country like India.Thirdly, as with any other observational study, there is always a potential for residual confounding factors despite the use of propensity

matching. Our results are based on the evaluation of two COVID-19 vaccines (COVISHIELD, COVAXIN), both of which were non-mRNA vaccines, being administered in India. Large scale studies involving different population group and vaccine types (both mRNA and non-mRNA vaccines) are required before the findings can be generalized to geographically diverse populations groups.

## Conclusion

In a large, populous and resource limited country such as India, the two indigenous vaccines with good immunogenic response have shown to decrease the mortality at 30 days and 6 months after AMI.

## Supporting information

**S1 Table. Comparison of patients' characteristics between 30 days outcome.**
(DOCX)

**S2 Table. Propensity score weighted logistic regression models for assessing the odds ratio of vaccination on mortality during one—six months of follow-up.**
(DOCX)

## Author Contributions

**Conceptualization:** Mohit D. Gupta, Shekhar Kunal, Girish M. P., Aarti Gupta, Vishal Batra, Ankit Bansal, Rakesh Yadav, Jamal Yusuf, Saibal Mukhopadhyay.

**Data curation:** Girish M. P., Dixit Goyal, Rajeev Kumar Malhotra, Prashant Mishra, Mansavi Shukla, Aarti Gupta, Vanshika Kohli, Ankit Bansal.

**Formal analysis:** Mohit D. Gupta, Rajeev Kumar Malhotra, Vanshika Kohli.

**Investigation:** Shekhar Kunal, Girish M. P., Dixit Goyal, Rajeev Kumar Malhotra, Vishal Batra, Ankit Bansal.

**Methodology:** Mohit D. Gupta, Shekhar Kunal, Girish M. P., Dixit Goyal, Aarti Gupta, Nitya Bundela, Vishal Batra, Ankit Bansal, Rakesh Yadav, Jamal Yusuf, Saibal Mukhopadhyay.

**Resources:** Shekhar Kunal.

**Software:** Shekhar Kunal.

**Supervision:** Mohit D. Gupta, Shekhar Kunal.

**Validation:** Shekhar Kunal.

**Writing – original draft:** Mohit D. Gupta, Shekhar Kunal, Girish M. P., Rajeev Kumar Malhotra, Prashant Mishra, Mansavi Shukla, Aarti Gupta, Vanshika Kohli, Nitya Bundela, Vishal Batra, Ankit Bansal, Rakesh Yadav, Jamal Yusuf, Saibal Mukhopadhyay.

**Writing – review & editing:** Mohit D. Gupta, Shekhar Kunal, Rajeev Kumar Malhotra, Prashant Mishra, Mansavi Shukla, Aarti Gupta, Vanshika Kohli, Nitya Bundela, Vishal Batra, Ankit Bansal, Rakesh Yadav, Jamal Yusuf, Saibal Mukhopadhyay.

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
