## [Decision Letter · Decision Letter 0]

3 May 2023

PONE-D-23-07327Impact of COVID-19 Vaccination on Mortality after Acute Myocardial InfarctionPLOS ONE

Dear Dr. Gupta,

Thank you for submitting your manuscript to PLOS ONE. After careful consideration, we feel that it has merit but does not fully meet PLOS ONE’s publication criteria as it currently stands. Therefore, we invite you to submit a revised version of the manuscript that addresses the points raised during the review process.

Please revise.

We look forward to receiving your revised manuscript.

Kind regards,

Academic Editor

PLOS ONE

Journal Requirements:

2. Please provide additional details regarding participant consent. In the ethics statement in the Methods and online submission information, please ensure that you have specified what type you obtained (for instance, written or verbal, and if verbal, how it was documented and witnessed). If your study included minors, state whether you obtained consent from parents or guardians. If the need for consent was waived by the ethics committee, please include this information

3. We note that you have stated that you will provide repository information for your data at acceptance. Should your manuscript be accepted for publication, we will hold it until you provide the relevant accession numbers or DOIs necessary to access your data. If you wish to make changes to your Data Availability statement, please describe these changes in your cover letter and we will update your Data Availability statement to reflect the information you provide

Reviewers' comments:

Reviewer's Responses to Questions

**Comments to the Author**

1. Is the manuscript technically sound, and do the data support the conclusions?

Reviewer #1: Yes

2. Has the statistical analysis been performed appropriately and rigorously? 

Reviewer #1: Yes

3. Have the authors made all data underlying the findings in their manuscript fully available?

Reviewer #1: Yes

4. Is the manuscript presented in an intelligible fashion and written in standard English?

Reviewer #1: Yes

5. Review Comments to the Author

Reviewer #1: At the outset, I would like to congratulate the authors for having taken up this study. In this study, the authors have evaluated two COVID-19 vaccines (COVISHIELD, COVAXIN) available in India. The primary outcome of the study was all-cause mortality both at one month and at six months of follow-up. The authors conclude that COVID-19 vaccines showed a decrease in all-cause mortality at 30 days and six months following AMI.

It is a very elegant study but it must be kept in mind that this is a single centre retrospective observational study.

A few questions for the authors:

• It is not mentioned how many participants received single dose and how many received two doses of vaccine.

• Why the effect of third dose (booster)of COVISHIELD not studied in this study?

• Why only one parameter that is all-cause mortality was evaluated in this study?

6. PLOS authors have the option to publish the peer review history of their article (what does this mean?). If published, this will include your full peer review and any attached files.

Reviewer #1: No

---

## [Author Response · Author response to Decision Letter 0]

10 May 2023

Comments to the Author

1. Is the manuscript technically sound, and do the data support the conclusions?

Reviewer #1: Yes

Reply: We thank the learned reviewer for the comments.

2. Has the statistical analysis been performed appropriately and rigorously?

Reviewer #1: Yes

Reply: We thank the learned reviewer for the comments.

3. Have the authors made all data underlying the findings in their manuscript fully available?

Reviewer #1: Yes

4. Is the manuscript presented in an intelligible fashion and written in standard English?

Reviewer #1: Yes

 Reply: We thank the learned reviewer for the comments.

5. Review Comments to the Author

Reviewer #1: At the outset, I would like to congratulate the authors for having taken up this study. In this study, the authors have evaluated two COVID-19 vaccines (COVISHIELD, COVAXIN) available in India. The primary outcome of the study was all-cause mortality both at one month and at six months of follow-up. The authors conclude that COVID-19 vaccines showed a decrease in all-cause mortality at 30 days and six months following AMI.

It is a very elegant study but it must be kept in mind that this is a single centre retrospective observational study.

A few questions for the authors:

• It is not mentioned how many participants received single dose and how many received two doses of vaccine.

Reply: We thank the learned reviewer for the comments. We have now highlighted in our MS regarding the number of participants receiving single and two doses of COVID-19 vaccine.

• Why the effect of third dose (booster)of COVISHIELD not studied in this study?

Reply: We thank the learned reviewer for the comment. The data included in this study was before the launch of the booster dose of COVID-19 vaccines in India and hence enrolled patients had not received the booster dose.

• Why only one parameter that is all-cause mortality was evaluated in this study?

Reply: We thank the learned reviewer for the comment. We had included all-cause mortality as one of the hard end-points for our study. In a resource limited developing country such as India during the COVID-19 pandemic, it was not possible to collect data regarding all other end points. This has been highlighted as one of the limitations of our study.

6. PLOS authors have the option to publish the peer review history of their article (what does this mean?). If published, this will include your full peer review and any attached files.

Do you want your identity to be public for this peer review? For information about this choice, including consent withdrawal, please see our Privacy Policy.

Reviewer #1: No

---

## [Decision Letter · Decision Letter 1]

21 Jun 2023

PONE-D-23-07327R1Impact of COVID-19 Vaccination on Mortality after Acute Myocardial InfarctionPLOS ONE

Dear Dr. Gupta,

Thank you for submitting your manuscript to PLOS ONE. After careful consideration, we feel that it has merit but does not fully meet PLOS ONE’s publication criteria as it currently stands. Therefore, we invite you to submit a revised version of the manuscript that addresses the points raised during the review process.

Please revise.

We look forward to receiving your revised manuscript.

Kind regards,

Academic Editor

PLOS ONE

Reviewers' comments:

Reviewer's Responses to Questions

**Comments to the Author**

1. If the authors have adequately addressed your comments raised in a previous round of review and you feel that this manuscript is now acceptable for publication, you may indicate that here to bypass the “Comments to the Author” section, enter your conflict of interest statement in the “Confidential to Editor” section, and submit your "Accept" recommendation.

Reviewer #2: All comments have been addressed

Reviewer #3: All comments have been addressed

Reviewer #4: (No Response)

2. Is the manuscript technically sound, and do the data support the conclusions?

Reviewer #2: Yes

Reviewer #3: No

Reviewer #4: (No Response)

3. Has the statistical analysis been performed appropriately and rigorously? 

Reviewer #2: Yes

Reviewer #3: No

Reviewer #4: (No Response)

4. Have the authors made all data underlying the findings in their manuscript fully available?

Reviewer #2: Yes

Reviewer #3: No

Reviewer #4: (No Response)

5. Is the manuscript presented in an intelligible fashion and written in standard English?

Reviewer #2: Yes

Reviewer #3: No

Reviewer #4: (No Response)

6. Review Comments to the Author

Reviewer #2: Very elegant manuscript. Well written and well structured, with a valid hypothesis. Conclusions should be interpreted with caution, with this only being a single centre study.

Reviewer #3: - missing base line characteristics as COPD, heart failure, CKD, are needed.

- no laboratory or imaging data were included.

- missing other variables as type of AMI (Stemi or NSTEMI) would affect the MACE

- the method and timing of revascularization would significantly affect the study results.

Reviewer #4: We have some comments:

1- The authors should defend the rationale on why including only STEMI ACS,

2- In the methodology and to have clear inclusion criteria, the authors should determine the maximum and minimum time interval between vaccination day and occurence of STEMI event needed for enrollment.

3- All cause mortality is a complex outcome with grand multifactorial confounders and is difficult to analyse. The authors should defend and discuss this point properly.

Regards

7. PLOS authors have the option to publish the peer review history of their article (what does this mean?). If published, this will include your full peer review and any attached files.

Reviewer #2: No

Reviewer #3: No

Reviewer #4: **Yes: **Rami Riziq Yousef Abumuaileq

---

## [Author Response · Author response to Decision Letter 1]

27 Jun 2023

Reviewers' comments:

Reviewer's Responses to Questions

Comments to the Author

1. If the authors have adequately addressed your comments raised in a previous round of review and you feel that this manuscript is now acceptable for publication, you may indicate that here to bypass the “Comments to the Author” section, enter your conflict of interest statement in the “Confidential to Editor” section, and submit your "Accept" recommendation.

Reviewer #2: All comments have been addressed

Reviewer #3: All comments have been addressed

Reviewer #4: (No Response)

Authors reply: We thank the learned reviewers for the comments.

2. Is the manuscript technically sound, and do the data support the conclusions?

Reviewer #2: Yes

Reviewer #3: No

Reviewer #4: (No Response)

Authors reply: We thank the learned reviewers for the comments.

3. Has the statistical analysis been performed appropriately and rigorously?

Reviewer #2: Yes

Reviewer #3: No

Reviewer #4: (No Response)

Authors reply: We thank the learned reviewers for the comments.

4. Have the authors made all data underlying the findings in their manuscript fully available?

Reviewer #2: Yes

Reviewer #3: No

Reviewer #4: (No Response)

Authors reply: We thank the learned reviewers for the comments.

5. Is the manuscript presented in an intelligible fashion and written in standard English?

Reviewer #2: Yes

Reviewer #3: No

Reviewer #4: (No Response)

Authors reply: We thank the learned reviewers for the comments.

6. Review Comments to the Author

Reviewer #2: Very elegant manuscript. Well written and well structured, with a valid hypothesis. Conclusions should be interpreted with caution, with this only being a single centre study.

Authors reply: We thank respected reviewer for kind words of appreciation. We agree that all the results have to be interpreted with caution. We have highlighted very clearly in the manuscript that this is a single centre study and results have to be interpreted with caution.

Reviewer #3: - missing base line characteristics as COPD, heart failure, CKD, are needed.

- no laboratory or imaging data were included.

Authors reply: We thank the learned reviewers for the comments. We have now added the data for missing baseline characteristics in the MS (Table 1). Since this was a retrospective review to determine any causality between COVID-19 vaccination and AMI, laboratory and imaging data were not included.

- missing other variables as type of AMI (Stemi or NSTEMI) would affect the MACE

Authors reply: We thank the learned reviewers for the comments. We had categorically looked into occurrence of STEMI following COVID-19 vaccination which we had mentioned in the methods. We sought to study the most severe form of MI after vaccination. Since this was a retrospective review carried out during the pandemic in a resource limited set-up, all comer STEMI cases were included.

- the method and timing of revascularization would significantly affect the study results.

Authors reply: We thank the learned reviewers for the comments. The details of revascularization with PCI have now been included in the manuscript. The figure (Figure 3) and table (table 1) have been modified. There was no impact of revascularization on the protective effect of COVID-19 vaccination. This has now been added in the MS.

Reviewer #4: We have some comments:

1- The authors should defend the rationale on why including only STEMI ACS,

Authors reply: We thank the learned reviewers for the comments. This study was a retrospective analysis of effect of vaccination on STEMI patients. This is the most severe form of coronary artery disease and hence we aimed to see the effect of vaccination on mortality after STEMI.

2- In the methodology and to have clear inclusion criteria, the authors should determine the maximum and minimum time interval between vaccination day and occurence of STEMI event needed for enrollment.

Authors reply: Learned Reviewer have very rightly suggested the mention of time interval. However, we must state that the effect of vaccination on STEMI and the time period after which it may or may not effect is unknown. Hence, we didn’t define any time interval in inclusion, rather we included all the patients who were vaccinated and studied the effect of vaccination in time quartiles to clearly see the time interval when these events occur.

3- All cause mortality is a complex outcome with grand multifactorial confounders and is difficult to analyse. The authors should defend and discuss this point properly.

Authors reply: Learned Reviewer has very rightly pointed out the limitation of using all cause mortality. However, we must state that all cause mortality is a hard end point and we have tried to adjust all the possible confounders in our analysis. In this study during covid times, it was extremely challenging to collect data of all possible outcomes in a resource limited country like India. The same has now been included in the discussion

---

## [Decision Letter · Decision Letter 2]

22 Aug 2023

Impact of COVID-19 Vaccination on Mortality after Acute Myocardial Infarction

PONE-D-23-07327R2

Dear Dr. Gupta,

We’re pleased to inform you that your manuscript has been judged scientifically suitable for publication and will be formally accepted for publication once it meets all outstanding technical requirements.

Kind regards,

Academic Editor

PLOS ONE

Additional Editor Comments (optional):

Reviewers' comments:

Reviewer's Responses to Questions

**Comments to the Author**

1. If the authors have adequately addressed your comments raised in a previous round of review and you feel that this manuscript is now acceptable for publication, you may indicate that here to bypass the “Comments to the Author” section, enter your conflict of interest statement in the “Confidential to Editor” section, and submit your "Accept" recommendation.

Reviewer #4: (No Response)

2. Is the manuscript technically sound, and do the data support the conclusions?

Reviewer #4: (No Response)

3. Has the statistical analysis been performed appropriately and rigorously? 

Reviewer #4: (No Response)

4. Have the authors made all data underlying the findings in their manuscript fully available?

Reviewer #4: (No Response)

5. Is the manuscript presented in an intelligible fashion and written in standard English?

Reviewer #4: (No Response)

6. Review Comments to the Author

Reviewer #4: (No Response)

7. PLOS authors have the option to publish the peer review history of their article (what does this mean?). If published, this will include your full peer review and any attached files.

Reviewer #4: **Yes: **Rami Riziq Yousef Abumuaileq

---

## [Editor Report · Acceptance letter]

25 Aug 2023

PONE-D-23-07327R2 

Impact of COVID-19 vaccination on mortality after acute Myocardial Infarction 

Dear Dr. Gupta:

I'm pleased to inform you that your manuscript has been deemed suitable for publication in PLOS ONE. Congratulations! Your manuscript is now with our production department. 

Kind regards, 

on behalf of

Dr. Robert Jeenchen Chen 

Academic Editor

PLOS ONE